# Brodmann Areas, V1 Atlas and Cognitive Impairment: Assessing Cortical Thickness for Cognitive Impairment Diagnostics

**DOI:** 10.3390/medicina60040587

**Published:** 2024-03-31

**Authors:** Maksims Trišins, Nauris Zdanovskis, Ardis Platkājis, Kristīne Šneidere, Andrejs Kostiks, Guntis Karelis, Ainārs Stepens

**Affiliations:** 1Department of Radiology, Riga Stradins University, LV-1007 Riga, Latvia; maksims.trisins@gmail.com (M.T.);; 2Department of Radiology, Riga East University Hospital, LV-1038 Riga, Latvia; 3Military Medicine Research and Study Centre, Riga Stradins University, LV-1007 Riga, Latvia; 4Department of Health Psychology and Pedagogy, Riga Stradins University, LV-1007 Riga, Latvia; 5Department of Neurology and Neurosurgery, Riga East University Hospital, LV-1038 Riga, Latviaguntis.karelis@rsu.lv (G.K.); 6Department of Infectology, Riga Stradins University, LV-1007 Riga, Latvia

**Keywords:** mild cognitive impairment, dementia, Alzheimer’s disease, cognition, Brodmann areas, cortical thickness, atlas-based segmentation, neuroimaging, structural MRI

## Abstract

*Background and Objectives*: Magnetic resonance imaging is vital for diagnosing cognitive decline. Brodmann areas (BA), distinct regions of the cerebral cortex categorized by cytoarchitectural variances, provide insights into cognitive function. This study aims to compare cortical thickness measurements across brain areas identified by BA mapping. We assessed these measurements among patients with and without cognitive impairment, and across groups categorized by cognitive performance levels using the Montreal Cognitive Assessment (MoCA) test. *Materials and Methods*: In this cross-sectional study, we included 64 patients who were divided in two ways: in two groups with (CI) or without (NCI) impaired cognitive function and in three groups with normal (NC), moderate (MPG) and low (LPG) cognitive performance according to MoCA scores. Scans with a 3T MRI scanner were carried out, and cortical thickness data was acquired using Freesurfer 7.2.0 software. *Results*: By analyzing differences between the NCI and CI groups cortical thickness of BA3a in left hemisphere (U = 241.000, *p* = 0.016), BA4a in right hemisphere (U = 269.000, *p* = 0.048) and BA28 in left hemisphere (U = 584.000, *p* = 0.005) showed significant differences. In the LPG, MPG and NC cortical thickness in BA3a in left hemisphere (H (2) = 6.268, *p* = 0.044), in V2 in right hemisphere (H (2) = 6.339, *p* = 0.042), in BA28 in left hemisphere (H (2) = 23.195, *p* < 0.001) and in BA28 in right hemisphere (H (2) = 10.015, *p* = 0.007) showed significant differences. *Conclusions*: Our study found that cortical thickness in specific Brodmann Areas—BA3a and BA28 in the left hemisphere, and BA4a in the right—differ significantly between NCI and CI groups. Significant differences were also observed in BA3a (left), V2 (right), and BA28 (both hemispheres) across LPG, MPG, NC groups. Despite a small sample size, these findings suggest cortical thickness measurements can serve as effective biomarkers for cognitive impairment diagnosis, warranting further validation with a larger cohort.

## 1. Introduction

Cognitive impairment refers to a decline in cognitive function that affects various cognitive domains and can impact daily functioning. Severe cognitive impairment is often considered equivalent to dementia, which is characterized by chronic cognitive decline in multiple domains leading to a significant decline in daily activities [1]. Dementia is not solely a normal part of aging but rather a condition that significantly affects individuals’ quality of life and independence. It is essential to distinguish between mild cognitive impairment, where cognitive function is impaired to a greater extent than is expected to be due to age, but the patient does not meet the criteria for dementia [2].

To diagnose cognitive impairment, especially in its early stages, a combination of clinical assessments and biomarkers plays a crucial role. Clinical examinations, including cognitive tests such as Montreal Cognitive Assessment (MoCA) and assessments of daily functioning, are typically used as the primary method for diagnosing cognitive impairment [3]. However, biomarkers can significantly aid in the early diagnosis of cognitive impairment, particularly in individuals without evident clinical signs of neurocognitive decline [4].

In the study by Battaglia et al. (2024), the focus was on the neural correlates and molecular mechanisms underlying memory and learning. The research emphasizes the critical role of memory and learning in acquiring, storing, and retrieving information essential for cognitive processes. This understanding is particularly relevant in the context of cognitive decline, where disruptions in memory and learning processes are often early indicators. By exploring the neural underpinnings and molecular pathways involved in memory and learning, this study contributes to a deeper understanding of how these cognitive functions are orchestrated at the neural level and how their impairment can lead to cognitive decline and neurodegenerative diseases such as Alzheimer’s [5].

MRI is a valuable tool in diagnosing cognitive impairment, particularly in conditions such as mild cognitive impairment (MCI) and Alzheimer’s disease (AD). Brodmann areas are distinct regions of the cerebral cortex identified based on cytoarchitectural variance. Mapping these areas using MRI can offer crucial insights into cognitive function and dysfunction [6].

The study by Battaglia et al. (2024) focuses on utilizing a multiscale integrated approach to investigate functional connectivity in fear learning. The research emphasizes the significance of functional brain connectivity assessments, particularly through techniques such as functional magnetic resonance imaging (fMRI). By employing fMRI, the study aims to gain valuable insights into how different brain regions interact and communicate during fear learning processes. Functional magnetic resonance imaging plays a crucial role in elucidating the neural mechanisms involved in fear learning, providing a deeper understanding of the functional connectivity patterns within the brain during these cognitive processes and could possibly aid in early diagnosis of cognitive decline [7].

Research has shown that MRI activation maps can be overlaid on automated anatomical labeling templates to estimate activation in Brodmann areas, aiding in visualizing cognitive processes and dysfunction [8]. Specifically, studies have indicated selective atrophy in Brodmann area 35 (BA35), approximating the trans-entorhinal region, in individuals with preclinical AD compared to controls, highlighting the utility of MRI in detecting early structural changes linked to cognitive decline [9]. Furthermore, alterations in cortical thickness can be indicative of cognitive decline, with specific regions such as the entorhinal cortex and precuneus playing crucial roles in cognitive impairment [10,11].

Alterations in cortical thickness can serve as indicators of cognitive decline due to the close relationship between brain structure and cognitive function. Changes in cortical thickness have been associated with various cognitive processes and disorders. For instance, studies have shown that alterations in cortical thickness in regions such as Brodmann Area 44, involved in language processing, and the primary motor cortex (BA4Ap) have shown changes in cortical thickness that may affect cognitive abilities [12,13]. Moreover, cortical thickness alterations in areas such as the entorhinal cortex (BA28) have been linked to memory deficits and cognitive decline, particularly in conditions such as Alzheimer’s disease [14]. Changes in cortical thickness in language-related regions, including Broca’s area (BA44) and peri-sylvian language networks, have been associated with language impairment and cognitive dysfunction [15,16]. Furthermore, structural connectivity deficits in regions such as the inferior frontal gyrus (BA44) have been observed in individuals with speech disorders, highlighting the role of cortical thickness in language function and cognitive processes [17].

This work aims to compare cortical thickness measurements in different areas of the brain divided using Brodmann area map parcellation in two ways: between the two groups—patients without cognitive impairment (NCI) and patients with impaired cognitive function (CI) according to MoCA, and also between the three groups (normal cognition (NC), moderate cognitive performance group (MPG) and low cognitive performance group (LPG)), which were divided based on the results of the MoCA test.

## 2. Materials and Methods

A total of 64 participants were enrolled in the cross-sectional study and then were divided into groups in two ways according to the Montreal Cognitive Assessment (MoCA) score:In two groups (with a score ≥ 26 and with a score <25);In three groups (with a score ≥ 26, with a score ≥ 15 and ≤25 and with a score ≤ 14).

Division in two groups. Patients without cognitive impairment (NCI) according to MoCA (with a score ≥ 26) and patients with impaired cognitive function (CI) according to MoCA (with a score < 25) [18].

Research participant demographic data, gender, and MoCA scores between the two groups can be seen in Table 1.

A Chi-square test on gender was performed, determining that there was a statistically significant difference between the groups (X^2^ = 4.512, *p* = 0.034). A Mann–Whitney U test did not find age differences in groups (U = 363.000, *p* = 0.584). MoCA scores were significantly different between the two groups (U = 799.000, *p* < 0.001).

Division in three groups. To better assess differences between severe cognitive impairment and mild/moderate cognitive impairment patients were divided as follows:Normal cognition group (NC)—participants with MoCA scores ≥ 26;Moderate cognitive performance group (MPG)—participants with MoCA ≥ 15 and ≤25;Low cognitive performance group (LPG)—participants with MoCA ≤ 14.

There were 17 participants in the NC group (mean age 70.941, SD 7.554, youngest 51 year, oldest 83 years. Mean MoCA score 27.529 (SD 1.231, lowest score 26, highest score 30).

MPG group contained 33 patients (mean age 72.758, SD 7.293, youngest 57 years, oldest 85 years). Mean MoCA score 22.121 (SD 2.792, lowest score 15, highest score 25).

LPG group contained 14 patients (mean age 74.286, SD 10.373, youngest 62, oldest 96). Mean MoCA score 8.786 (SD 3.786, lowest score 4, highest score 14). Research participant demographic data, gender, and MoCA scores between the three groups can be seen in Table 2.

A Chi-square test on gender was performed, determining that there were no statistically significant differences between the groups (X^2^ = 4.631, *p* = 0.099). Similarly, Kruskall–Wallis tests did not find age differences in groups (H (2) = 0.340, *p* = 0.844). MoCA scores were significantly different between the three groups (H (2) = 52.795, *p* < 0.001).

### 2.1. Selection of Participants

In our study, individuals were referred to a board-certified neurologist specializing in cognitive impairment for evaluation due to either subjective cognitive complaints or suspected cognitive decline. All of the selected patients had right hand dominance. Exclusion criteria encompassed clinically significant neurological conditions such as tumors, major strokes, intracerebral lobar hemorrhages, malformations, Parkinson’s disease, and multiple sclerosis, as well as substance, alcohol abuse, major depression, schizophrenia, and other psychiatric conditions. Additionally, patients with documented significant vascular diseases were not considered for inclusion. Magnetic resonance imaging (MRI) did not reveal any other noteworthy pathological findings in the study participants.

### 2.2. MRI Acquisition Protocol

MRI imaging was conducted using a 3.0 tesla MRI scanner within the setting of a university hospital. 3D T1 Ax (flip angle 11, TE min full, TI 400, FOV 25.6, layer thickness 1 mm) images were used for post-processing. Additional sequences were used to rule out other clinically significant pathologies, including, T2, 3D FLAIR, DWI, ADC, SWI sequences. 

### 2.3. Cortical Parcellation

Cortical reconstruction was performed by using Freesurfer 7.2.0 image analysis software. It is documented and freely available for download online (http://surfer.nmr.mgh.harvard.edu/ accessed on 16 February 2024). The technical details of these procedures are described in prior publications [19,20,21,22,23,24,25,26,27,28,29,30,31,32,33,34].

We used the Brodmann Area Maps (BA Maps) and Hinds V1 Atlas labelling protocol to extract cortical thickness results [35,36,37,38,39,40].

### 2.4. Statistical Analysis

JASP 0.18.3 was used for the statistical analysis (Eric-Jan Wagenmakers, Amsterdam, The Netherlands) [41]. Statistical analysis included descriptive statistics, a Chi-square test, a Mann–Whitney U test, a Kruskal–Wallis test, and Dunn’s post hoc analysis of study results.

Descriptive statistics were used to estimate general variables and differences between groups. A Chi-square test was used to determine the association between categorical variables. A Mann–Whitney U test was used to evaluate statistically significant differences between the patients without cognitive impairment and patients with impaired cognitive function according to MoCA. A Kruskall–Wallis test was used to evaluate statistically significant differences between the NC, MPG and LPG groups, and if there were statistically significant differences, Dunn’s post hoc test was utilized with additional Bonferroni and Holm corrections.

## 3. Results

### 3.1. Statistical Differences between Two Groups (NCI and CI)

By analyzing differences between NCI and CI groups using Mann-Whitney U test, statistically significant results were found in cortex thickness of primary somatosensory area (BA3a) in left hemisphere, primary anterior motor area (BA4a) in the right hemisphere and entorhinal cortex (BA28) in the left hemisphere (*p* < 0.05). Other regions did not show statistically significant differences between the NCI and CI groups, although some regions had a *p*-value close to statistically significant: primary somatosensory cortex (BA1) in the right hemisphere (*p* = 0.057), secondary visual cortex (V2) in the right hemisphere (*p* = 0.060) and entorhinal cortex (BA28) in the right hemisphere (*p* = 0.062). When presenting the *p* value, multiple comparison corrections were not made. Therefore, the results serve as exploratory data to be validated by a larger cohort and further multiple comparison corrections. The results are presented in Table 3.

### 3.2. Statistical Differences between Three Groups (LPG, MPG and NC)

We performed Kruskal-Wallis test to determine whether there are statistically significant differences between LPG, MPG and NC.

We did not find statistically significant differences in:Somatosensory area (BA1)—left hemisphere (H (2) = 1.762, *p* = 0.414), right hemisphere (H (2) = 4.718, *p* = 0.095);Somatosensory area (BA2)—left hemisphere (H (2) = 2.594, *p* = 0.273), right hemisphere (H (2) = 1.662, *p* = 0.436);Somatosensory area (BA3b)—left hemisphere (H (2) = 1.342, *p* = 0.511), right hemisphere (H (2) = 0.071, *p* = 0.965);Primary anterior motor area (B4a)—left hemisphere (H (2) = 1.721, *p* = 0.423), right hemisphere (H (2) = 4.075, *p* = 0.130);Primary posterior motor area (B4p)—left hemisphere (H (2) = 1.544, *p* = 0.462), right hemisphere (H (2) = 4.207, *p* = 0.122);Pre-motor area (B6)—left hemisphere (H (2) = 2.626, *p* = 0.269), right hemisphere (H (2) = 2.665, *p* = 0.264);Broca’s area pars opercularis (BA44)—left hemisphere (H (2) = 0.644, *p* = 0.725), right hemisphere (H (2) = 3.412, *p* = 0.182);Broca’s area pars triangularis (BA45)—left hemisphere (H (2) = 0.359, *p* = 0.836), right hemisphere (H (2) = 3.505, *p* = 0.173);Primary visual area (V1)—left hemisphere (H (2) = 3.438, *p* = 0.179), right hemisphere (H (2) = 4.164, *p* = 0.125);Middle temporal area (MT)—left hemisphere (H (2) = 0.152, *p* = 0.927), right hemisphere (H (2) = 0.818, *p* = 0.664);Perirhinal cortex (BA35)—left hemisphere (H (2) = 4.850, *p* = 0.088), right hemisphere (H (2) = 3.995, *p* = 0.136);Secondary visual area (V2)—left hemisphere (H (2) = 4.261, *p* = 0.119);Primary somatosensory area (B3a)—right hemisphere (H (2) = 0.404, *p* = 0.817).

#### 3.2.1. Primary Somatosensory Area Left Hemisphere—BA3a

The Kruskal–Wallis tests revealed a significant difference between groups evaluating cortical thickness in primary somatosensory area (BA3a) in the left hemisphere (H (2) = 6.268, *p* = 0.044) (see Figure 1).

By performing Dunn’s post hoc test, statistically significant differences were found between the MPG—NC group (*p* = 0.012, after Bonferroni and Holm correction statistical significance was maintained) (see Table 4).

#### 3.2.2. Secondary Visual Area Right Hemisphere—V2

The Kruskal–Wallis tests revealed a significant difference between groups evaluating cortical thickness in secondary visual area (V2) in the right hemisphere (H (2) = 6.339, *p* = 0.042) (see Figure 2).

By performing Dunn’s post hoc test, statistically significant differences were found between the MPG—NC group (*p* = 0.020, but after Bonferroni and Holm correction there were no statistically significant difference) (see Table 5).

#### 3.2.3. Entorhinal Cortex Left Hemisphere—BA28

The Kruskal–Wallis tests revealed a significant difference between groups evaluating cortical thickness in entorhinal cortex (BA28) in the left hemisphere (H (2) = 23.195, *p* < 0.001) (see Figure 3).

By performing Dunn’s post hoc test, statistically significant differences were found between the LPG—MPG group (*p* < 0.001, after Bonferroni and Holm correction statistical significance was maintained) and the LPG—NC group (*p* < 0.001, after Bonferroni and Holm correction statistical significance was maintained) (see Table 6).

#### 3.2.4. Entorhinal Cortex Right Hemisphere—BA28

The Kruskal–Wallis tests revealed a significant difference between groups evaluating cortical thickness in entorhinal cortex (BA28) in the right hemisphere (H (2) = 10.015, *p* = 0.007) (see Figure 4).

By performing Dunn’s post hoc test, statistically significant differences were found between the LPG—MPG group (*p* = 0.011, after Bonferroni and Holm correction statistical significance was maintained) and the LPG—NC group (*p* = 0.002, after Bonferroni and Holm correction statistical significance was maintained) (see Table 7).

## 4. Discussion

Till today scientists all around the world are trying to find early markers of dementia and cognitive impairment. Cortical thickness measurements have emerged as a valuable biomarker for cognitive impairment in various neurological conditions. In our study we tried to find connections between cortical thickness of different brain regions according to Brodmann area map and cognitive impairment. In discussion we compare our results with other research papers and provide possible explanations for our findings, as well as future directions.

BA4a is a primary anterior motor area responsible for motor planning and execution. Changes in cortical thickness in BA4a may influence motor control and coordination, which can impact cognitive functions related to motor skills and coordination [42,43]. Our study showed statistically significant results in NCI and CI groups in BA4a in the right hemisphere. One study suggests that cortical thickness alterations in BA4a and other motor areas may be associated with motor deficits and cognitive impairment in individuals with Parkinson’s disease [44].

BA3a is a primary somatosensory area involved in processing sensory information. Alterations in cortical thickness in BA3a may affect sensory perception, potentially impacting cognitive functions related to sensory processing and discrimination [45,46]. Our study showed statistically significant results between LPG, MPG, and NC groups and in the NCI and CI groups in BA3a in the left hemisphere. The results of one study suggest that alterations in cortical thickness in BA3b, a primary somatosensory area, may impact texture discrimination and perception, potentially contributing to cognitive impairment in individuals with these deficits [47].

V2 is the secondary visual cortex essential for visual processing and perception. It is involved in higher-order visual processing tasks such as color perception, motion detection, and object recognition. The connection between V2 and cognitive functioning is significant as V2 contributes to the integration and interpretation of visual stimuli, which are essential for various cognitive processes. It receives the information from V1 and process the information to other parts of the cortex [48]. Alterations in cortical thickness in V2 may affect visual perception and recognition, potentially impacting cognitive functions related to visual information processing and object recognition [49]. Our study showed statistically significant results between LPG, MPG, and NC groups in V2 in the right hemisphere. Additionally, it showed nearly significant results in the NCI and CI groups. Previously association between cortical thickness alterations in V2 region and cognitive impairment was observed individuals with Huntington’s disease [50].

The most promising region in a role of a biomarker proved to be entorhinal cortex (BA28) in both hemispheres, since it showed statistically significant results between LPG, MPG, and NC groups on both sides. Additionally, entorhinal cortex in the left hemisphere showed statistically significant results in the NCI and CI groups and in the right hemisphere showed nearly significant result between two groups. The entorhinal cortex (BA28) in both the left and right hemispheres plays a crucial role in spatial memory and episodic memory functions [51]. It is involved in navigation, formation, and consolidation of spatial and declarative memory [52]. The left entorhinal cortex is associated with object recognition and familiarity-based judgments [53]. Changes in the right entorhinal cortex have been associated with memory deficits and cognitive decline, highlighting its importance in cognitive function [54]. Additionally, the entorhinal cortex is correlated with episodic memory and is impacted by Alzheimer’s disease-related pathology [55]. The results reinforce studies that linked entorhinal cortex thickness to cognitive impairment [56,57,58].

Similar studies collectively emphasize the importance of investigating cognitive functions, brain volumes, metacognitive abilities, and white matter integrity in individuals with cognitive impairment, particularly in the context of mathematical knowledge, financial capacity, and related cognitive functions in conditions such as amnestic mild cognitive impairment (aMCI) and Alzheimer’s disease.

The study by Giannouli and Tsolaki (2023) focused on investigating brain volumes and metacognitive deficits in individuals with amnestic Mild Cognitive Impairment (aMCI) compared to healthy controls. They found positive correlations between knowledge of strategies (avoidance strategies) and amygdala on both sides and white matter volume [59]. In a study by Stoeckel et al. (2013), the researchers explored the relationship between the MRI volume of the medial frontal cortex and financial capacity in individuals with mild Alzheimer’s disease. The study indicated that the volume of the medial frontal cortex could predict financial capacity in patients with mild Alzheimer’s disease [60]. Giannouli and Tsolaki (2019) found a strong correlation between financial capacity in aMCI patients and the volumes of the right amygdala and left angular gyrus [61]. Gerstenecker et al. (2017) conducted a study focusing on the association between white matter degradation and reduced financial capacity in individuals with mild cognitive impairment (MCI) and Alzheimer’s disease (AD). The research highlighted that in AD, increased mean, and axial diffusivities in specific brain regions such as the anterior cingulate, callosum, and frontal areas were linked to poorer financial capacity [62].

By analyzing our data, we did not find significant differences in the NCI and CI groups, and similarly in the LPG, MPG, and NC groups apart from the structures mentioned above. This study was exploratory rather than confirmatory and performed on a small cohort. When presenting the *p* value, multiple comparison corrections were not made. Therefore, the results serve as exploratory data to be validated by a larger cohort and further multiple comparison corrections.

One of the primary limitations of this study is the relatively small sample size of 64 patients. A larger sample would provide more robust and generalizable findings. Another possible limitation is the cross-sectional design of the study. Longitudinal studies would offer more insights into the progression and predictive value of cortical thickness changes over time. Also, depression is known to impact cognitive functions and could potentially confound the relationship between cortical thickness measurements and cognitive impairment. The study does not account for the potential effects of depression cognitive-neuropsychological, and emotional characteristics of the participants on cognitive performance and cortical thickness. In the end, while some differences in cortical thickness reached statistical significance, the clinical significance of these findings in terms of actual cognitive impairment and functional outcomes remains to be determined.

Future studies should integrate multimodal imaging techniques such as functional MRI (fMRI) and diffusion tensor imaging (DTI) to examine the functional and structural connectivity changes related to cortical thickness alterations. Additionally, it would be beneficial to compare cortical thickness measurements from BA mapping with other cognitive assessment tools to assess the reliability and consistency of the results. Furthermore, research should focus on investigating the specific roles of the identified Brodmann areas (BA3a, BA4a, BA28, V2) in cognitive processes and include comprehensive cognitive-neuropsychological and emotional assessments to understand their influence on cortical thickness measurements and cognitive performance.

Utilizing cortical thickness measurements as biomarkers can aid in early diagnosis and intervention for cognitive impairment, improving the accuracy of diagnosis and allowing for personalized treatment plans. Additionally, identifying specific Brodmann areas linked to cognitive impairment can guide the development of educational and counseling programs to enhance cognitive function and quality of life for those experiencing cognitive decline.

## 5. Conclusions

In our study, cortical thickness of primary somatosensory area (BA3a) in left hemisphere, primary anterior motor area (BA4a) in the right hemisphere and entorhinal cortex (BA28) in the left hemisphere showed significant differences in the NCI and CI groups. Additionally, primary somatosensory area (BA3a) in the left hemisphere, secondary visual area (V2) in the right hemisphere and entorhinal cortex (BA28) in both hemispheres showed significant differences in the LPG, MPG, and NC groups. Post-hoc tests showed significant differences in MPG—NC groups in primary somatosensory area (BA3a) in the left hemisphere and in LPG—MPG together with LPG—NC groups in entorhinal cortex (BA28) of both hemispheres.

Despite the limitations we believe our study provides the foundation for a new way of using cortical thickness measurements as quantitative biomarkers in diagnosis of cognitive impairment and dementia.

Further analysis with a larger patient cohort is needed to evaluate and validate the diagnostic certainty and prognostic value of cortical thickness measurement in different areas of the brain using Brodmann area map parcellation.

## Figures and Tables

**Figure 1 medicina-60-00587-f001:**
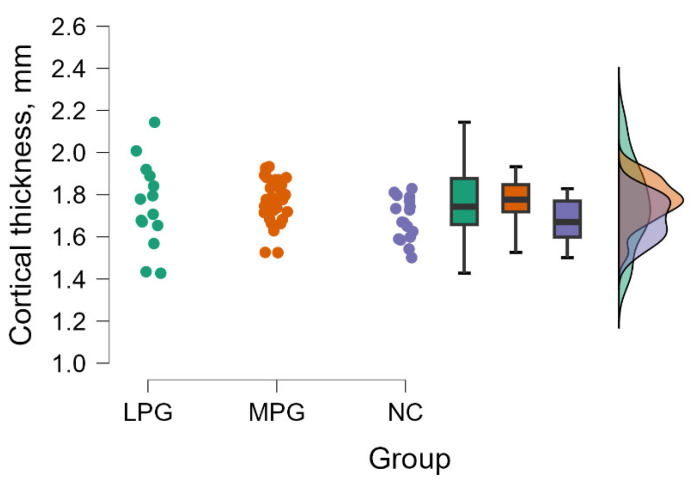
Cortical thickness comparisons in primary somatosensory area (BA3a) in the left hemisphere between (from left to right) LPG, MPG, and NC with data distribution in each group.

**Figure 2 medicina-60-00587-f002:**
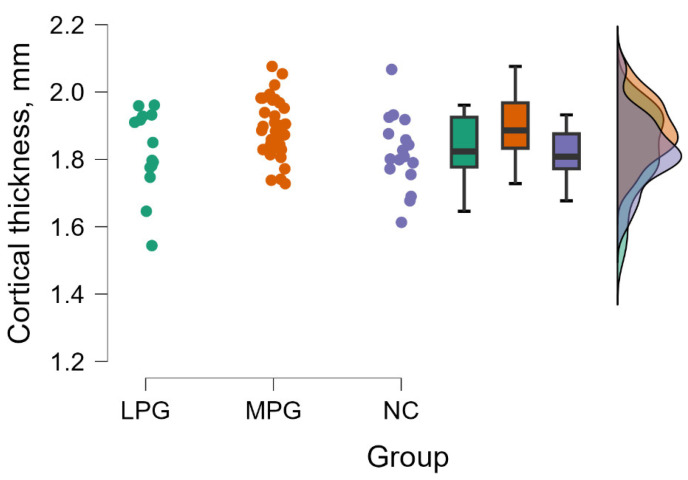
Cortical thickness comparisons in secondary visual area (V2) in the right hemisphere between (from left to right) LPG, MPG, and NC with data distribution in each group.

**Figure 3 medicina-60-00587-f003:**
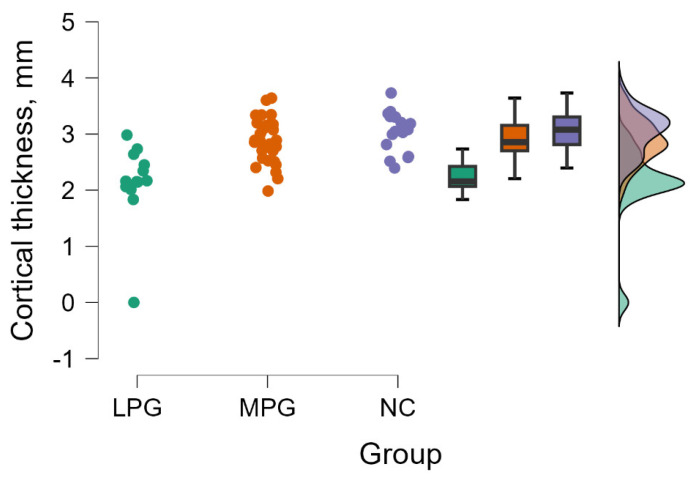
Cortical thickness comparisons in entorhinal cortex (BA28) in the left hemisphere between (from left to right) LPG, MPG, and NC with data distribution in each group.

**Figure 4 medicina-60-00587-f004:**
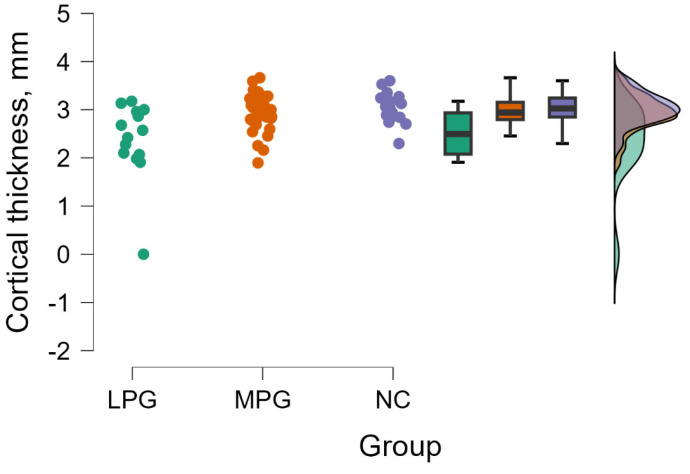
Cortical thickness comparisons in entorhinal cortex (BA28) in the right hemisphere between (from left to right) LPG, MPG, and NC with data distribution in each group.

**Table 1 medicina-60-00587-t001:** Demographic data and Montreal Cognitive Assessment (MoCA) scores in study patients (divided in three groups).

	Gender (F:M)		Age		MoCA	
	NCI	CI	NCI	CI	NCI	CI
Valid	16:1	32:15	17	47	17	47
Mean			70.941	73.213	27.529	18.149
Std. Deviation			7.554	8.241	1.231	6.890
Minimum			51.000	57.000	26.000	4.000
Maximum			83.000	96.000	30.000	25.000

**Table 2 medicina-60-00587-t002:** Demographic data and Montreal Cognitive Assessment (MoCA) scores in study patients (divided in three groups).

	Gender (F:M)			Age			MoCA		
	LPG	MPG	NC	LPG	MPG	NC	LPG	MPG	NC
Valid	10:4	22:11	16:1	14	33	17	14	33	17
Mean				74.286	72.758	70.941	8.786	22.121	27.529
Std. Deviation				10.373	7.293	7.554	3.786	2.792	1.231
Minimum				62.000	57.000	51.000	4.000	15.000	26.000
Maximum				96.000	85.000	83.000	14.000	25.000	30.000
X^2^		4.631							
H (2)					0.340			52.795 ***	

LPG—low cognitive performance group; MPG—moderate cognitive performance group; NC—normal cognition group. *** = *p* < 0.001.

**Table 3 medicina-60-00587-t003:** The Mann–Whitney U test by comparing the NCI and CI patient groups.

	W	df	*p*
BA1 in the left hemisphere	340.000		0.373
BA1 in the right hemisphere	274.000		0.057
BA2 in the left hemisphere	370.000		0.659
BA2 in the right hemisphere	333.000		0.316
BA3a in the left hemisphere	241.000		0.016 *
BA3a in the right hemisphere	377.500		0.744
BA3b in the left hemisphere	373.000		0.693
BA3b in the right hemisphere	408.500		0.897
BA4a in the left hemisphere	314.000		0.196
BA4a in the right hemisphere	269.000		0.048 *
BA4p in the left hemisphere	349.000		0.451
BA4p in the right hemisphere	280.500		0.072
BA6 in the left hemisphere	309.500		0.174
BA6 in the right hemisphere	299.000		0.128
BA44 in the left hemisphere	348.000		0.438
BA44 in the right hemisphere	303.500		0.147
BA45 in the left hemisphere	360.500		0.558
BA45 in the right hemisphere	287.000		0.089
V1 in the left hemisphere	404.000		0.952
V1 in the right hemisphere	397.500		0.982
V2 in the left hemisphere	310.500		0.179
V2 in the right hemisphere	275.500		0.060
MT in the left hemisphere	415.000		0.820
MT in the right hemisphere	359.500		0.548
BA35 in the left hemisphere	351.500		0.470
BA35 in the right hemisphere	483.000		0.209
BA28 in the left hemisphere	584.000		0.005 *
BA28 in the right hemisphere	523.000		0.062

* *p* < 0.05.

**Table 4 medicina-60-00587-t004:** Dunn’s post hoc comparison of the MoCA score and cortical thickness in primary somatosensory area (BA3a) in the left hemisphere between the LPG—MPG, LPG—NC, and MPG—NC groups.

Comparison	z	W_i_	W_j_	*p*	p_bonf_	p_holm_
LPG—MPG	−0.680	33.036	37.076	0.496	1.000	0.496
LPG—NC	1.467	33.036	23.176	0.142	0.427	0.285
MPG—NC	2.501	37.076	23.176	0.012	0.037	0.037

**Table 5 medicina-60-00587-t005:** Dunn’s post hoc comparison of the MoCA score and cortical thickness in secondary visual area (V2) in the right hemisphere between the LPG—MPG, LPG—NC, and MPG—NC groups.

Comparison	z	W_i_	W_j_	*p*	p_bonf_	p_holm_
LPG—MPG	−1.669	28.179	38.091	0.095	0.285	0.190
LPG—NC	0.442	28.179	25.206	0.658	1.000	0.658
MPG—NC	2.318	38.091	25.206	0.020	0.061	0.061

**Table 6 medicina-60-00587-t006:** Dunn’s post hoc comparison of the MoCA score and cortical thickness entorhinal cortex (BA28) in the left hemisphere between the LPG—MPG, LPG—NC, and MPG—NC groups.

Comparison	z	W_i_	W_j_	*p*	p_bonf_	p_holm_
LPG—MPG	−3.915	12.250	35.500	<0.001	<0.001	<0.001
LPG—NC	−4.629	12.250	43.353	<0.001	<0.001	<0.001
MPG—NC	−1.413	35.500	43.353	0.158	0.473	0.158

**Table 7 medicina-60-00587-t007:** Dunn’s post hoc comparison of the MoCA score and cortical thickness entorhinal cortex (BA28) in the right hemisphere between the LPG—MPG, LPG—NC, and MPG—NC groups.

Comparison	z	W_i_	W_j_	*p*	p_bonf_	p_holm_
LPG—MPG	−2.548	19.250	34.379	0.011	0.033	0.022
LPG—NC	−3.053	19.250	39.765	0.002	0.007	0.007
MPG—NC	−0.969	34.379	39.765	0.333	0.998	0.333

## Data Availability

The datasets presented in this article are not readily available since they could have potentially identifiable data. Requests to access the datasets should be directed to nauris.zdanovskis@rsu.lv.

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
