# Peer review of "Brodmann Areas, V1 Atlas and Cognitive Impairment: Assessing Cortical Thickness for Cognitive Impairment Diagnostics"

_medicina, 2024, doi:10.3390/medicina60040587_

Round 1

Reviewer 1 Report

Comments and Suggestions for Authors

The paper titled " Brodmann Areas, V1 Atlas and Cognitive Impairment: Assessing Cortical Thickness for Cognitive Impairment Diagnostics" by Trišins et al., explores the relationship between cortical thickness measurements in different brain areas identified by Brodmann area mapping and cognitive impairment. The study included 64 participants divided into groups based on cognitive performance levels assessed using the Montreal Cognitive Assessment (MoCA) test. Significant differences in cortical thickness were found between groups with and without cognitive impairment, particularly in specific Brodmann areas like BA3a, BA4a, and BA28. The study highlights the potential of using cortical thickness measurements as biomarkers for diagnosing cognitive impairment, emphasizing the need for further research with larger cohorts to validate these findings. Overall, the study explores for the first time the possibility of utilizing cortical thickness measurements as diagnostic biomarkers for cognitive impairment, with implications for early detection and treatment strategies in conditions like mild cognitive impairment and Alzheimer's disease.

In general, I think the idea of this article is really interesting and the authors’ fascinating observations on this timely topic may be of interest to the readers of Medicina. However, some comments, as well as some crucial evidence that should be included to support the author’s argumentation, needed to be addressed to improve the quality of the manuscript, its adequacy, and its readability prior to the publication in the present form. My overall judgment is to publish this paper after the authors have carefully considered my suggestions below, in particular reshaping parts of the ‘Introduction’ and ‘Methods’ sections by adding more evidence.

Please consider the following comments: 

The title effectively conveys the study's focus on cortical thickness assessment for diagnosing cognitive impairment using Brodmann Areas and V1 Atlas. However, it could be improved by being more concise and direct to capture the essence of the research.

A graphical abstract that will visually summarize the main findings of the manuscript is highly recommended.

The abstract provides a comprehensive overview of the study's objectives, methods, results, and conclusions. Still, in my opinion it could benefit from including more specific details about the key findings and implications of the study to provide a clearer snapshot of the research.

The introduction effectively defines cognitive impairment, dementia, and mild cognitive impairment, setting a solid foundation for the study. In this section, authors could provide a more detailed discussion on the specific Brodmann areas and their relevance to cognitive function and further explore existing literature on cortical thickness measurements in relation to cognitive decline, to strengthen the scientific background of the study [1-2]. Additionally, I would suggest incorporating more recent studies or advancements in MRI techniques related to cortical thickness measurements and cognitive impairment, to better demonstrate the study's alignment with current research trends.

The methodology section is detailed in explaining participant selection criteria, MRI acquisition protocol, cortical parcellation using Freesurfer software, and statistical analysis methods. However, the authors did not clearly explain how the groups were divided based on MoCA scores. Please, provide a more detailed explanation of the group division criteria, to ensure the reproducibility of the study.

The discussion section interprets the results well by linking cortical thickness measurements to cognitive function and emphasizing their significance as potential biomarkers for diagnosing cognitive impairment. However, I would ask the authors to add a more in-depth analysis of the limitations of the study, suggestions for future research directions, and practical implications of the findings in clinical settings.

In summary, while the paper provides valuable insights into cortical thickness measurements for diagnosing cognitive impairment, improvements in clarity, conciseness, engagement, and depth of analysis in certain sections could enhance its overall impact and readability.

I hope that, after careful revisions, the manuscript can meet the journal’s high standards for publication. I declare no conflict of interest regarding this manuscript.

Best regards,

Reviewer

References:

1. https://doi.org/10.3390/ijms25020864

2. https://doi.org/10.3390/ijms25052724

Comments on the Quality of English Language

Minor editing of English language required.

Author Response

Dear reviewer, 

Thank you very much for taking the time to review our manuscript. We appreciate Your insights and please find the detailed responses below:

“The title effectively conveys the study's focus on cortical thickness assessment for diagnosing cognitive impairment using Brodmann Areas and V1 Atlas. However, it could be improved by being more concise and direct to capture the essence of the research.”

Thank you for your valuable feedback. While we appreciate your suggestion, we would prefer to retain the current title as it effectively conveys the study's focus on cortical thickness assessment for diagnosing cognitive impairment using Brodmann Areas and V1 Atlas.

“A graphical abstract that will visually summarize the main findings of the manuscript is highly recommended.”

We have made a graphical abstract as you suggested. It can be seen in the new version of our paper.

“The abstract provides a comprehensive overview of the study's objectives, methods, results, and conclusions. Still, in my opinion it could benefit from including more specific details about the key findings and implications of the study to provide a clearer snapshot of the research.”

We have rewritten the conclusion parts to better emphasise our findings.

“The introduction effectively defines cognitive impairment, dementia, and mild cognitive impairment, setting a solid foundation for the study. In this section, authors could provide a more detailed discussion on the specific Brodmann areas and their relevance to cognitive function and further explore existing literature on cortical thickness measurements in relation to cognitive decline, to strengthen the scientific background of the study [1-2]. Additionally, I would suggest incorporating more recent studies or advancements in MRI techniques related to cortical thickness measurements and cognitive impairment, to better demonstrate the study's alignment with current research trends.”

We investigated the studies you provided to us and decided to add them to the introduction section of our research as they serve as valuable examples of current research trends.

“The methodology section is detailed in explaining participant selection criteria, MRI acquisition protocol, cortical parcellation using Freesurfer software, and statistical analysis methods. However, the authors did not clearly explain how the groups were divided based on MoCA scores. Please, provide a more detailed explanation of the group division criteria, to ensure the reproducibility of the study.”

Thank You for the suggestion! We have rewritten the methodology section to better explain group division and be more concise.

“The discussion section interprets the results well by linking cortical thickness measurements to cognitive function and emphasizing their significance as potential biomarkers for diagnosing cognitive impairment. However, I would ask the authors to add a more in-depth analysis of the limitations of the study, suggestions for future research directions, and practical implications of the findings in clinical settings.”

Thank You for the suggestion! We added more limitations of the study and provided our vision on future research directions, and practical implications of the findings in clinical settings in the new version of a publication.

Thank you for your time and suggestions!

Have a nice day!

Best regards,

Maksims Trišins

Reviewer 2 Report

Comments and Suggestions for Authors

This is a well written research article focusing on the possible correlation of cortical thickness in several brain areas with the cognitive performance.

The introduction gives the background of this study. It includes the definition of dementia and cognitive impairment, the role of MRI and the possible importance of cortical thickness in several Broadmann areas.

The methodology section describes the population of the study, the inclusion criteria implemented, the MRI protocol and the statistical analyses used during this study.

The results are interesting enough and thoroughly described. However, I think that some figures depicting the Broadmann areas could further enhance the quality of this paper.

In the discussion, the authors comments on the results of the study and mention relative literature data. I think that a paragraph, summarizing the main limitation of the study should be added.

The conclusion section summarizes the findings of the current study in a adequate way.

Comments on the Quality of English Language

English language is well used and the style seems to be appropriate.

Author Response

Dear reviewer, 

Thank you very much for taking the time to review our manuscript. We appreciate Your insights and please find the detailed responses below:

“I think that some figures depicting the Brodmann areas could further enhance the quality of this paper.”

We agree that visual information will only improve our publication. We added a graphical abstract to enhance the quality of our paper.

“I think that a paragraph summarising the main limitation of the study should be added.”

We added a paragraph describing main limitations of the study in the discussion section.

Thank you for your time and suggestions!

Have a nice day! 

Best regards, 

Maksims Trišins

Reviewer 3 Report

Comments and Suggestions for Authors

This study tries to correlate MRI cortical thickness and cognitive function.  Patiient's hand dominancy was not considered. Study limitation was not discussed.

Author Response

Dear reviewer, 

Thank you very much for taking the time to review our manuscript. We appreciate Your insights and please find the detailed responses below:

Patient's hand dominancy was not considered.

We agree that this information is necessary to consider. As all of our patients had right hand dominance, we added additional information about hand dominance in section “2.1. Selection of participants.”

Thank you for your time and suggestions!

Have a nice day! 

Best regards, 

Maksims Trišins

Reviewer 4 Report

Comments and Suggestions for Authors

I do not understand figures 1 and 2. Figure 1 clearly shows that average cortical thickness was greater in the low and moderate groups than in the normals. The same is seen in Figure 2. Please explain why cognitive impairment was associated with increased cortical thickness.

Author Response

Dear reviewer, 

Thank you very much for taking the time to review our manuscript. We appreciate Your insights and please find the detailed responses below:

“Figure 1 clearly shows that average cortical thickness was greater in the low and moderate groups than in the normals. The same is seen in Figure 2. Please explain why cognitive impairment was associated with increased cortical thickness.”

Our suggestion is that in both cases, the increased cortical thickness could be a result of compensatory mechanisms, neuroplasticity, pathological changes, or alterations in functional connectivity that are associated with cognitive impairment. The brain's attempt to adapt and compensate for cognitive deficits might lead to structural changes, including increased cortical thickness in the primary somatosensory area of the left hemisphere (BA3a) and the secondary visual area of the right hemisphere (V2).

For example, in patients with multiple sclerosis, increased cortical thickness in visual sensory regions was identified, suggesting adaptive plasticity [1]. Similarly, in patients with schizophrenia, altered cortical thickness in various brain regions was found, with some areas showing increased thickness, potentially as a compensatory factor [2].

We believe that the relationship between cortical thickness and cognitive impairment is complex, and further research would be needed to fully understand the exact mechanisms underlying our observations.

References:

  1. https://doi.org/10.1007/s00429-012-0440-3
  2. https://doi.org/10.1093/schbul/sbt177

Thank you for your time and suggestions!

Have a nice day! 

Best regards, 

Maksims Trišins

Reviewer 5 Report

Comments and Suggestions for Authors

The topic of this article is of interest, but there are some points that need revision:

Why was MoCA used instead of other similar tests, such as MMSE? Please justify.

The recruitment process and the demographic, cognitive-neuropsychological, and emotional characteristics of the participants are not included. Were participants checked for mood disorders? If yes, with what tests? 

How was the diagnosis for cognitive deficits reached? Please explain in detail in the Methods section.

Table 2 is difficult to follow. Please present not acronyms, but full names.

Similar studies have found that arithmetic/financial abilities/capacity are linked to changes in specific brain areas (volumes) in neurocognitive disorders in the first stages and these can serve as biomarkers in early diagnosis of cognitive impairment. For exaxmple, strong correlations in MCI patients were found between diminishing financial abilities and diminished left angular gyrus and amygdala volumes (see and discusss https://pubmed.ncbi.nlm.nih.gov/30877733/) as well as white matter degradation (doi: 10.3233/JAD-170341).This also holds true for older patients with mild AD ( doi: 10.1007/s11682-013-9226-3). Also longitutidal data in MCI patients shows the importance of brain changes and metacognitive deficits regarding arithmetic/financial tasks over time as another early marker of cognitive deterioration in older adults, please read and discuss: doi: 10.3390/diagnostics13040680

The first paragraph in the Discussion section is not needed, but more relevant references must be added so conclusions can be compared to other already published research on biomarkers.

Comments on the Quality of English Language

Moderate English language editing.

Author Response

Dear reviewer, 

Thank you very much for taking the time to review our manuscript. We appreciate Your insights and please find the detailed responses below: 

“Why was MoCA used instead of other similar tests, such as MMSE? Please justify.”

The MoCA scale is well-suited for the Latvian population and is extensively utilized in hospitals throughout the country. It has also been employed in previous research, as shown in the following study: https://doi.org/10.2174/1567202616666190129153409. This research highlighted the use of the MoCA scale in Latvian or Russian language versions to evaluate cognition, underscoring its relevance in this population. In contrast, the MMSE is not as commonly used, which is why we opted for the MoCA scale.

“The recruitment process and the demographic, cognitive-neuropsychological, and emotional characteristics of the participants are not included. Were participants checked for mood disorders? If yes, with what tests?”

Specific patient recruitment was not conducted. In our study, individuals were referred to a board-certified neurologist specializing in cognitive impairment for evaluation due to either subjective cognitive complaints or suspected cognitive decline. Exclusion criteria encompassed clinically significant neurological conditions which included major depression, schizophrenia, and other psychiatric conditions (we added this to the “2.1. Selection of Participants”). There was no specific screening for mood disorders, but if there were clinically significant pathologies - patients were not recruited.

“How was the diagnosis for cognitive deficits reached? Please explain in detail in the Methods section.”

In our study, we exclusively utilized the MoCA scale to evaluate the cognition of the participants. Therefore, we refrain from using the term "cognitive deficit" and instead use the term "cognitive performance group." Given the absence of standardized criteria for group categorization, apart from a MoCA score of ≥ 26 indicating no cognitive impairment, we chose to categorize our patients in two distinct ways:

  • In two groups – patients without cognitive impairment (NCI) according to MoCA (with a score ≥ 26) and patients with impaired cognitive function (CI) according to MoCA (with a score <25).
  • In three groups - normal cognition group (NC) - participants with MoCA scores ≥ 26 , moderate cognitive performance group (MPG) - participants with MoCA ≥ 15 and ≤25 and low cognitive performance group (LPG) - participants with MoCA ≤ 14.

We have also rewritten the methodology section to better explain group division and be more concise.

“Table 2 is difficult to follow. Please present not acronyms, but full names.”

We have added acronym explanations below the Table 2 to improve its readability.

“The first paragraph in the Discussion section is not needed.”

As you recommend removing the first paragraph of the discussion, we believe it provides a concise overview of the current situation in the world and serves as an introduction to the subsequent publications being discussed.

“Similar studies have found that arithmetic/financial abilities/capacity are linked to changes in specific brain areas (volumes) in neurocognitive disorders in the first stages and these can serve as biomarkers in early diagnosis of cognitive impairment.”

“… more relevant references must be added so conclusions can be compared to other already published research on biomarkers.”

We investigated the studies you provided to us and decided to add them to the discussion part of our research as they serve as relevant data on possible biomarkers.

Thank you for your time and suggestions!

Have a nice day! 

Best regards, 

Maksims Trišins

Round 2

Reviewer 1 Report

Comments and Suggestions for Authors

Dear Authors,

I am pleased to acknowledge that you have indeed addressed all of my concerns and queries in a clear and precise manner. Your responses have provided valuable insights into the modifications made to the manuscript in light of my comments. It is evident that you have taken great care to ensure that the revised manuscript aligns more closely with the scientific rigor expected for publication in Medicina. Having reviewed the revised manuscript, I am satisfied with the changes that have been implemented.

In light of the above, I am pleased to recommend acceptance of your manuscript for publication in Medicina.

Best regards,

Reviewer

Reviewer 3 Report

Comments and Suggestions for Authors

I recognized significant improvement of this manuscript.

Reviewer 4 Report

Comments and Suggestions for Authors

Thank you for clarifying the postulated relationship between cortical thickness and cognition.